# Thrombotic Risk and Coagulation Imbalance in Cirrhosis and Hepatocellular Carcinoma: Clinical Implications and Management

**DOI:** 10.3390/cancers17213413

**Published:** 2025-10-23

**Authors:** Leonardo Stella, Matteo De Siati, Rosa Talerico, Maria Pallozzi, Lucia Cerrito, Silvia Sorrentino, Antonio Gasbarrini, Erica De Candia, Roberto Pola, Francesca Romana Ponziani

**Affiliations:** 1Liver Unit, CEMAD Centro Malattie del’Apparato Digerente, Medicina Interna e Gastroenterologia, Fondazione Policlinico Universitario A. Gemelli IRCCS, 00168 Rome, Italy; maria.pallozzi@guest.policlinicogemelli.it (M.P.); lucia.cerrito@policlinicogemelli.it (L.C.); antonio.gasbarrini@unicatt.it (A.G.); francesca.ponziani@gmail.com (F.R.P.); 2Unit of Haemostasis and Thrombosis Diseases, Fondazione Policlinico Universitario A. Gemelli IRCCS, 00168 Rome, Italy; m.desiati97@gmail.com (M.D.S.); silvia.sorrentino@policlinicogemelli.it (S.S.); erica.decandia@policlinicogemelli.it (E.D.C.); 3Department of Aging, Orthopedic, and Rheumatologic Sciences, Fondazione Policlinico Universitario A. Gemelli IRCCS, Università Cattolica del Sacro Cuore, 00168 Rome, Italy; rosa.talerico@guest.policlinicogemelli.it (R.T.); roberto.pola@unicatt.it (R.P.); 4Department of Translational Medicine and Surgery, Università Cattolica del Sacro Cuore, 00168 Rome, Italy

**Keywords:** hepatocellular carcinoma, liver cancer, cirrhosis, hemostasis, thrombosis, portal vein thrombosis, venous thromboembolism, immunotherapy, systemic therapy, anticoagulation

## Abstract

**Simple Summary:**

Hepatocellular carcinoma (HCC) disrupts hemostasis, increasing risks of thrombosis and bleeding. Cirrhosis impairs coagulation through reduced factor synthesis, endothelial dysfunction, and inflammation, while tumor-driven factors, such as tissue factor expression, platelet activation, and reduced fibrinolysis, enhance hypercoagulability. Advanced assays like thrombin generation (TGA) and rotational thromboelastometry (ROTEM) improve coagulation assessment and risk stratification. Anticoagulation benefits selected patients with portal vein thrombosis (PVT) or venous thromboembolism (VTE) when liver function is preserved and may be safely used prophylactically in surgical cases. In immunotherapy, especially with anti-VEGF agents, bleeding risk must be carefully managed. Individualized anticoagulation guided by functional testing may safely reduce thrombosis and improve outcomes in HCC.

**Abstract:**

Hepatocellular carcinoma (HCC) is characterized by a complex disruption of hemostatic balance, increasing the risk of both thrombotic and hemorrhagic events. Thrombotic complications, most notably portal vein thrombosis (PVT) and venous thromboembolism (VTE), have a significant impact on clinical outcomes and therapeutic strategies. Cirrhosis contributes to the precarious equilibrium between pro- and anticoagulant forces through impaired synthesis of coagulation factors, endothelial dysfunction, and systemic inflammation. In the presence of HCC tumor-driven mechanisms, such as tissue factor expression, extracellular vesicle release, platelet activation, and suppression of fibrinolysis exacerbate this prothrombotic state. In this scenario, advanced diagnostic tools such as thrombin generation assay (TGA) and rotational thromboelastometry (ROTEM) offer a more accurate assessment of coagulation dynamics than conventional tests, enabling better risk stratification especially for therapeutic purposes. Anticoagulant therapy has demonstrated clinical benefit in selected cases of non-malignant PVT and VTE, particularly when liver function is preserved. While prophylactic strategies are still under investigation, data suggest they may be safely implemented in selected surgical patients. In the setting of immunotherapy, especially regimens involving anti-VEGF agents, anticoagulation may be considered with careful management of bleeding risk due to portal hypertension. An individualized approach to anticoagulation, supported by functional coagulation testing, is gaining acceptance as a means to safely reduce thrombotic burden and potentially improve outcomes in patients with HCC.

## 1. Background

Hepatocellular carcinoma (HCC) is the most common primary malignancy of the liver and ranks among the top five causes of cancer-related mortality worldwide [1]. HCC typically arises in chronic liver disease, with cirrhosis present in approximately 90% of cases [2]. Cirrhosis leads to a delicate imbalance between procoagulant and anticoagulant factors, thereby increasing the risk of both bleeding and thrombotic complications [3]. This dual vulnerability underscores not only the oncogenic potential of chronic liver injury but also its role in predisposing patients to vascular events.

Venous thromboembolism (VTE) is a well-recognized complication in cancer patients, with the highest risk occurring within the first 3 to 6 months after cancer diagnosis; in some cases, VTE may even precede the detection of malignancy. VTE is associated with a worsened prognosis and represents a substantial healthcare burden, with a fatal recurrence rate reaching up to 15% [4,5]. In the context of HCC, cancer-associated thrombosis is a significant concern, with patients exhibiting a 3- to 7-fold higher risk of VTE compared to the general population [6,7].

Portal vein thrombosis (PVT) is the most common thrombotic complication observed in patients with cirrhosis [8]. Its pathogenesis is multifactorial, involving factors such as reduced portal blood flow, endothelial dysfunction, systemic inflammation, and a hypercoagulable state. The presence of HCC further exacerbates this prothrombotic milieu, thereby amplifying the thrombotic risk.

The management of HCC is particularly challenging due to the intricate interplay between tumor biology and the profoundly altered hemostatic profile characteristic of advanced liver disease. Hepatic dysfunction compromises the synthesis of both procoagulant and anticoagulant factors [9,10], resulting in a fragile hemostatic balance. This concurrent predisposition to bleeding and thrombosis complicates therapeutic strategies and significantly impacts clinical outcomes. To optimize clinical decision-making, the integration of dynamic diagnostic tools such as thrombin generation assay (TGA) and rotational thromboelastometry (ROTEM) may offer a more precise evaluation of thrombotic and hemorrhagic risk compared to traditional coagulation parameters.

## 2. Incidence and Risk Factors

### 2.1. Portal Vein Thrombosis

The incidence of PVT in patients with HCC shows marked variability across the literature, with reported prevalence rates ranging from approximately 10% to nearly 50% [11,12,13,14,15,16,17,18,19,20,21,22,23,24,25,26,27,28,29] (Figure 1). Despite this variability, a consistent pattern emerges across study designs and populations: patients with HCC are at a significantly higher risk of PVT compared to those with cirrhosis alone (Table 1).

Retrospective cohort studies have been instrumental in characterizing the burden of PVT in HCC. In the liver transplant setting, earlier studies have demonstrated that PVT is significantly more prevalent in patients with concomitant HCC compared to cirrhotic controls. For instance, Ravaioli et al. [13] reported a PVT prevalence of 40.7% in cirrhotic patients with concomitant HCC, compared to 30.7% in those without HCC, while Serag [24] reported PVT prevalence of 22.7% in cirrhotic patients with HCC versus 12.7% in those without. These findings were corroborated by Connolly et al. [15], who observed a 31% prevalence of PVT in a mixed cohort of HCC patients undergoing liver-directed therapies or transplantation. PVT was strongly associated with advanced disease characteristics and poor survival outcomes. Malaguarnera et al. [21] further emphasized the thrombotic burden, reporting PVT in nearly half (49.2%) of their HCC cohort in acute setting.

Prospective studies, although fewer in number, reinforce these observations and provide longitudinal data on thrombotic risk. In a well-characterized cohort, Zanetto et al. [18] reported a one-year cumulative incidence of PVT of 24.4% among patients with HCC, a significantly higher rate compared to cirrhotic controls (11.4%). This increased risk was associated with hypercoagulability parameters such as elevated clot firmness. Violi et al. [22], while not finding a direct association between HCC and incident PVT after adjustment, found that prior PVT and thrombocytopenia were dominant predictors of thrombotic recurrence, highlighting the chronic and relapsing nature of PVT in advanced liver disease. Siddiqui et al. [29], focusing specifically on patients with early-stage cirrhosis (Child–Pugh A), found a 24.5% prevalence of PVT, emphasizing that even compensated patients with large tumors and elevated alpha-fetoprotein (AFP) levels are at significant thrombotic risk.

Population-based and registry studies further contextualize these findings on a broader scale. Faccia et al. [28] evaluated over 7000 cirrhotic patients admitted to a large third-level Italian Hospital based on international classification of diseases (ICD) codes, and they identified a PVT prevalence of 10.6% among those with HCC. Larger data from Ponziani et al. [17], with 23,932 cirrhotic transplant candidates (39% with HCC), reported an overall PVT prevalence of 10.6%, with new cases occurring at a rate of 3.2 per 100 person-years and a total prevalence of 7.4 per 100 person-years.

Risk factors for PVT in the context of HCC are multifactorial and can be grouped into distinct clinical domains. Portal hypertension is a major contributor: the presence of clinically significant portal hypertension (CSPH), including large varices, reduced portal flow, portosystemic collaterals, and prior sclerotherapy or splenectomy, has been consistently associated with the development of PVT. CSPH was significantly associated with PVT in patients undergoing liver transplantation or locoregional therapies, and high-risk patients often exhibiting a small liver or large-volume portosystemic shunting [11,25,27].

Liver dysfunction is another independent risk factor for PVT. PVT was significantly more frequent in patients with advanced liver disease, including those with Child–Pugh class B/C, hypoalbuminemia, elevated international normalized ratio (INR), or hepatic encephalopathy. These associations have been confirmed in both transplant and non-transplant cohorts, with Child–Pugh class B/C and low albumin identified as independent predictors of thrombosis [15,20,21,22].

Tumor-related factors, particularly tumor burden and biological aggressiveness, also play a central role. Large tumor size (>5.9 cm), high alpha-fetoprotein (AFP > 40–400 ng/mL), and major vascular invasion were independently associated with PVT in multiple studies. Tumors that are progressive or non-responsive to treatment are more likely to develop thrombosis, and the lack of response to HCC treatment has been identified as a predictor of both PVT and mortality [15,27,29].

A prior history of PVT or splanchnic thrombosis significantly increases risk of recurrence, and it was one of the strongest predictors of new events in prospective cohorts [22,25]. Furthermore, hypercoagulability parameters, such as elevated clot firmness (FIBTEM MCF > 25 mm), have been linked to a higher risk of PVT, particularly in patients with early-stage compensated HCC [18].

### 2.2. Venous Thromboembolism

The reported prevalence of VTE in patients with HCC ranges between 1% and 6% [15,16,18,20,23,26,28,30,33,34,35] (Figure 1), with variation depending on population characteristics, clinical setting, and detection methods (Table 1). In a retrospective cohort of cirrhotic patients with HCC, Connolly et al. [15] observed a VTE prevalence of 6.7%, which increased to 11.5% among patients with concurrent PVT. Similarly, Wang et al. [20] reported a 5.9% incidence of VTE in a cohort of 270 cirrhotic patients with HCC, with PE accounting for 43.8%, DVT for 25%, and intra-abdominal thrombosis for 37.5% of cases. In the emergency department setting, Faccia et al. [28] found a VTE prevalence of 1.7% among patients with cirrhosis and HCC, compared to 1.3% in cirrhotic patients without HCC. Among patients undergoing liver surgery for HCC, the VTE incidence was reported as 2.88% [16].

Larger administrative and population-based studies confirm the presence of VTE risk in broader HCC populations. Al-Taee et al. [23], using a national inpatient database, observed a VTE prevalence of 2.8% among over 54,000 hospitalized patients with HCC, with a statistically significant increase from 2008 to 2013. In surgical populations, Schlick et al. [33] reported an overall VTE rate of 3.1% in more than 11,000 patients undergoing hepatectomy for malignancy, with 0.9% of cases occurring post-discharge. Finally, in critically ill patients, Ow et al. [26] identified early VTE in 13% of cirrhotic ICU patients, 80% of which were PVT, but with a significant proportion represented by DVT or PE.

Multiple risk factors for systemic thromboembolism in patients with HCC have been identified across clinical settings. Hepatic dysfunction remains central: Child–Pugh B cirrhosis was significantly associated with VTE in both the Connolly and Wang cohorts. Similarly, hepatic encephalopathy and liver failure were identified as predictors of VTE in emergency and surgical settings [28,33].

Tumor burden and metastatic disease also play a key role. Wang et al. [20] found that multifocal intrahepatic lesions and extrahepatic metastases were independently associated with VTE. Al-Taee et al. [23] further identified metastatic spread as a strong predictor of VTE in hospitalized patients, along with older age and African American ethnicity.

Metabolic comorbidities such as obesity and diabetes mellitus contribute to thrombotic risk. Obesity has been consistently identified as a risk factor across both medical and surgical studies [20,33], while Lesmana et al. [30] found diabetes to be an independent predictor of DVT in cirrhotic patients (OR 4.26).

In perioperative settings, procedural factors such as extent of liver resection, renal insufficiency, perioperative transfusion, and non-surgical interventions have been independently associated with postoperative VTE [33]. Finally, in the intensive care unit (ICU) setting, factors such as sepsis, elevated bilirubin levels, and cryoprecipitate administration were significant predictors of early thromboembolic events in cirrhotic patients, including those with HCC [26].

## 3. Mechanisms of Hemostatic Alterations

Hemostatic alterations in cirrhosis and HCC result from a delicate and unstable balance between procoagulant and anticoagulant forces (Figure 2). These mechanisms collectively determine the risk of both thrombosis and bleeding, significantly influencing disease progression and clinical outcomes.

### 3.1. Cirrhosis-Related Alterations

Cirrhosis, once viewed predominantly as a bleeding disorder, is now recognized as a condition characterized by rebalanced but unstable hemostasis, in which both procoagulant and anticoagulant pathways are concurrently impaired [36]. This fragile equilibrium is highly sensitive to systemic or vascular insults, which can shift the balance toward either bleeding or thrombosis. Notably, TGA often reveal preserved or even enhanced thrombin production, suggesting a paradoxical prothrombotic shift despite impaired hepatic synthesis of clotting factors [36].

This hemostatic instability results from a complex interplay of pathophysiological mechanisms. Reduced levels of liver-derived procoagulant factors (II, V, VII, IX, X, XI, XII and XIII) are counterbalanced by deficiencies in endogenous anticoagulants such as protein C, protein S, and antithrombin III [37]. Compounding this, the fibrinolytic system is frequently dysregulated. Many patients exhibit hypofibrinolysis, characterized by elevated levels of plasminogen activator inhibitor-1 (PAI-1), which impairs clot degradation and promotes thrombus formation [38]. Conversely, a minority may experience hyperfibrinolysis, associated with increased bleeding risk [39]. Additionally, elevated levels of von Willebrand factor (vWF) and factor VIII, both primarily of endothelial origin, contribute to a hypercoagulable state, particularly in the presence of thrombocytopenia and qualitative platelet dysfunction [9,40,41]. Despite reduced platelet counts, typically due to splenic sequestration and decreased thrombopoietin production, platelet adhesion remains functional owing to vWF overexpression [42]. Circulating activated platelets and reduced platelet lifespan further complicate to this altered hemostasis.

Portal hypertension, a hallmark of cirrhosis, exacerbates this prothrombotic milieu by promoting venous stasis, endothelial dysfunction, and low-grade systemic inflammation, which are core elements of Virchow’s triad [36]. Decreased portal vein flow velocity predisposes patients to PVT [43], while intrahepatic microthrombosis may contribute to parenchymal extinction, fibrosis progression [44], and worsening liver function.

An additional contributor is systemic endothelial dysfunction, as reflected by increased levels of adhesion molecules such as ICAM-1 and VCAM-1, which perpetuate vascular inflammation and thrombotic activation [45,46]. Cirrhosis induces capillarization and loss of nitric oxide (NO) production due to GRK2 upregulation and oxidative stress, impairing endothelial relaxation and enhancing intrahepatic vascular resistance [40]. Platelet-derived TGF-β1 plays a pivotal role in cirrhosis-related hypercoagulability and endothelial dysfunction. TGF-β1 levels were significantly higher in patients with PVT (6866.55 vs. 3840.60 pg/mL; *p* = 0.015) and were strongly correlated with platelet count (r = 0.733; *p* < 0.001) and thromboelastography parameters. In vitro, TGF-β1 stimulation of liver sinusoidal endothelial cells increased expression of vWF, thrombomodulin, ICAM-1, and VEGF, confirming its prothrombotic and endothelial-injuring effects [47]. Furthermore, the circulation of prothrombotic microvesicles, particularly those expressing tissue factor and derived from endothelial cells, platelets, and leukocytes, amplifies systemic coagulation activation [31].

Other systemic factors further destabilize this precarious balance. Malnutrition, particularly protein–calorie deficiency, is common in advanced cirrhosis and, along with vitamin K deficiency due to malabsorption or cholestasis, impairs the synthesis of both pro- and anticoagulant proteins [48]. Renal dysfunction alters platelet function by disrupting thromboxane metabolism and calcium signaling [49], while infections may provoke the release of endogenous heparinoid-like substances, heightening the risk of bleeding in select contexts [50].

Finally, genetic predispositions such as the G20210A prothrombin gene mutation have been associated with a higher incidence of PVT, particularly in patients with decompensated liver disease [51].

### 3.2. Tumor-Related Factors

Recent integrative transcriptomic analyses have identified two coagulation-related molecular subtypes of HCC. The procoagulant/immunosuppressive subtype exhibits heightened expression of prothrombotic genes, an immune-excluded tumor microenvironment (TME), and is correlated with adverse clinical outcomes. In contrast, the immunologically active subtype, enriched in CD8^+^ T-cell infiltration, is associated with improved survival [52].

A central driver of the prothrombotic phenotype is the overexpression of tissue factor (TF) on the surface of HCC cells. TF initiates the extrinsic coagulation cascade, triggering thrombin generation and fibrin deposition [53], events that are strongly linked to vascular invasion and poor prognosis [54]. Concurrently, TF-bearing tumor cells activate protease-activated receptors (PARs) on sinusoidal endothelial cells, potentiating local thrombus formation in the portal venous system [45]. This procoagulant shift is further amplified under hypoxia, as HIF 1α stabilization induces TF and PAI 1, while oncogenic YAP–TEAD4 signaling transcriptionally upregulates PAI 1, collectively suppressing fibrinolysis and reinforcing tumor-associated thrombosis.

Extracellular vesicles (EVs), particularly TF-expressing microvesicles (MVs) derived from tumor cells, platelets, leukocytes, and endothelial cells, serve as circulating procoagulant platforms. These vesicles expose phosphatidylserine and TF, providing catalytic surfaces for coagulation complex assembly. Circulating TF-positive MVs are significantly elevated in HCC, particularly in presence of coexisting cirrhosis, and strongly correlate with PVT [31].

The platelet axis further contributes via tumor-derived thrombopoietin (TPO), which is overexpressed in a subset of patients with paraneoplastic thrombocytosis (3–9%). High TPO levels are associated with increased tumor burden, high AFP, PVT, and poorer prognosis. In contrast, surgical resection leads to a reduction in TPO and platelet counts, supporting the tumor-driven nature of platelet activation [55].

Another hallmark of HCC vascular biology is the contribution of cancer stem cells (CSCs), which express EpCAM and CD133 and are strongly implicated in vascular invasion [56]. CSCs exhibit enhanced adhesion and trans-endothelial migration, likely facilitated by ICAM 1 and VCAM 1 upregulation on sinusoidal endothelium. In preclinical models, CSC-enriched inocula generate more intra portal tumor emboli, providing direct evidence of the link between stemness traits and vascular occlusion [57].

Similarly, epithelial-to-mesenchymal transition (EMT) facilitates the release of circulating tumor cells (CTCs), which may trigger thrombosis either as embolic elements or through paracrine procoagulant signals. In vivo, murine models injected with CSQT 2 HCC cells develop portal thrombosis, confirming the prothrombotic potential of EMT-associated CTCs. However, whether direct clot initiation or microenvironment modulation predominates remains under investigation [58].

The tumor microenvironment also promotes thrombosis through protease activity. Urokinase-type plasminogen activator (uPA) and matrix metalloproteinases (MMPs) remodel extracellular matrix, exposing procoagulant phospholipids and TF [59,60]. Concurrently, the YAP/TEAD4–PAI 1 axis and HIF 1α signaling create a hypofibrinolytic milieu, synergizing with tumor-driven platelet activation to favor thrombus persistence [61,62].

At the post-transcriptional level, microRNAs integrate the thrombo-inflammatory network. miR 122, a liver-specific tumor suppressor, is frequently downregulated in HCC. Its loss enhances tumor proliferation, migration, and EMT, driven by IGF 1R/PI3K/AKT and Wnt/β catenin signaling, and is mechanistically linked to increased thrombotic risk, including PVT [63,64].

### 3.3. Platelet-Related Factors

In both cirrhotic and non-cirrhotic HCC, platelet hyperreactivity is a key driver of thrombotic risk. Aggregometry studies using ADP, arachidonic acid, or thrombin demonstrate that patients with hyperactivated platelets are significantly more likely to develop PVT [65]. This prothrombotic tendency is potentiated by elevated von Willebrand factor, which enhances platelet aggregation and thrombus formation [65].

HCC cells may also contribute via thrombopoietin (TPO) secretion, with 3–9% of patients demonstrating thrombocytosis, although splenic sequestration in cirrhosis may mask this effect [66]. In cirrhosis, splenic sequestration from portal hypertension may mask this effect [67]. Beyond its role in hemostasis, platelets actively support tumor progression by activating hepatic stellate cells, endothelial cells, and tumor-associated macrophages, thereby promoting fibrosis, angiogenesis, and immune evasion [68]. Defective primary hemostasis, characterized by platelet plug formation disproportionate to thrombocytopenia, reflects in vivo platelet activation. This is supported by the correlation between urinary thromboxane metabolite excretion and plasma prothrombin fragment 1 + 2, suggesting that coagulation cascade activation reinforces platelet hyperreactivity.

P selectin, expressed by activated platelets and endothelial cells, is both a mediator and biomarker of cancer associated thrombosis. Plasma soluble P selectin (sP selectin) levels greater than ~53 ng/mL confer a 2.6-fold increased risk of VTE, with a 6 month incidence of 11.9% compared to 3.7% in patients below this threshold [69]. Clinical studies report mean sP selectin levels of 33.6 ± 23.4 ng/mL in patients with thrombosis versus 20.4 ± 6.9 ng/mL in those without, with 21–22 ng/mL serving as a practical discriminative cutoff [69]. Elevated sP selectin is also observed in HCV-related chronic hepatitis, cirrhosis, and HBV/HCV-related HCC, correlating with viral load, disease stage, and thrombosis [70].

A growing body of evidence highlights the prothrombotic and pro-tumor roles of platelet-derived extracellular vesicles (pEVs). pEVs expressing CD61 are linked to thrombotic risk in viral cirrhosis, suggesting that platelet activation is integral to the procoagulant cascade [71]. Surface markers such as CD61 and CD62P mediate interactions with the endothelium and immune cells, while phosphatidylserine and tissue factor provide procoagulant platforms that accelerate thrombin generation and fibrin deposition. Within the tumor microenvironment, pEVs deliver VEGF, PDGF, and pro inflammatory cytokines (IL 6, IL 8), promoting angiogenesis, immune evasion, and fibrosis through M2 macrophage polarization. Additionally, pEVs transfer microRNAs (miR 223, miR 126) and mitochondrial material, driving tumor metabolic reprogramming and invasiveness [72].

At the post transcriptional level, miR 126 is a pivotal regulator of platelet activation, vascular homeostasis, and thrombosis. Highly enriched in platelets and endothelial progenitor cells (EPCs), miR 126 modulates aggregation, secretion, and integrin signaling. Murine models lacking intraplatelet miR 126 exhibit impaired thrombus formation and prolonged bleeding times. Mechanistically, miR 126 enhances platelet activation by targeting PIK3R2, the regulatory subunit of PI3K, thereby activating the AKT pathway. In EPCs, miR 126 promotes angiogenesis and thrombus resolution via PI3K/AKT, Ras/ERK, and SDF 1/CXCR7 pathways, while enhancing pro angiogenic and anti-inflammatory cytokine production. Loss of miR 126 function may therefore contribute to vascular dysfunction, delayed thrombus resolution, and persistent hypercoagulability in HCC [73,74].

### 3.4. Coagulation and Fibrinolytic Factors

In HCC, hypercoagulability is a central driver of PVT, arising from a complex interplay of increased procoagulant activity, impaired anticoagulant defenses, and dysregulated fibrinolysis.

Among procoagulant factors, fibrinogen is consistently elevated, with thromboelastographic studies demonstrating increased clot firmness in HCC patients. Notably, even in the absence of cirrhosis, HCC alone is sufficient to trigger systemic coagulation activation. Non-cirrhotic patients with HCC frequently present with elevated fibrinogen levels and hyperactive platelets, which correlate with tumor burden and systemic inflammation. Both of these factors are independently associated with thrombotic events and adverse prognosis [18,65]. This underscores the intrinsic thrombogenicity of the tumor itself, independent of the underlying liver architecture. Elevated fibrinogen has been linked to poorer survival and enhanced metastatic potential, highlighting its dual role as both a biomarker and mediator of tumor progression [75].

Factor VIII (FVIII) is also elevated in HCC, reflecting the underlying hypercoagulable state and predicting thromboembolic complications. In a cohort of 30 patients, those with VTE had a median FVIII levels of 2.325 IU/mL (IQR 1.65–2.77) compared to 1.641 IU/mL (IQR 0.77–2.70) in patients without VTE, while healthy controls had an average of 1.197 IU/mL (IQR 0.86–1.89). High FVIII, particularly when combined with soluble P selectin or CD62 expression, identified patients at greatest risk, with 33.3% of these patients developing VTE [76]. Similarly, von Willebrand factor (vWF) is markedly elevated in HCC, reflecting endothelial activation and tumor-associated hypercoagulability. High vWF:Ag levels correlate with larger tumor burden, advanced fibrosis, and portal hypertension. Preoperative vWF:Ag levels above 291% are associated with worse overall survival (median 39.8 vs. 73.4 months) and an increased risk of post-hepatectomy liver failure [77]. In patients with decompensated cirrhosis, ultra-large vWF multimers promote platelet aggregation and PVT, linking vWF to tumor progression via vascular and microthrombotic mechanisms [77].

In contrast, endogenous anticoagulant defenses are frequently impaired in HCC. ADAMTS13 activity is often reduced, leading to the accumulation of ultra large vWF multimers and sinusoidal microthrombosis [78,79]. In cirrhotic HCC patients, a low ADAMTS13/vWF:GPIbR ratio (<0.4) predicts non-tumoral PVT and correlates with poorer outcomes [42].

Antithrombin III (ATIII) levels are also diminished in HCC, reflecting both impaired hepatic synthesis and a prothrombotic state. In cirrhotic patients with PVT—40% of whom had HCC—plasma ATIII levels < 54% were associated with significantly worse 5-year survival (HR 3.68; 95% CI 1.66–8.16; *p* = 0.0013). Low ATIII was an independent predictor of liver failure-related mortality, even when HCC-related deaths were excluded, and retained superior prognostic value compared to Child–Pugh and ALBI scores [80].

Profound fibrinolytic dysregulation further favors thrombus persistence. Hepatic-dependent factors, including plasminogen and factor XIII, are significantly reduced in both HCC and cirrhosis with PVT, resulting in decreased plasmin generation and suboptimal fibrin cross linking [81]. Conversely, circulating tissue type plasminogen activator (tPA) is markedly elevated, primarily due to reduced hepatic clearance, and is accompanied by increased plasminogen activator inhibitor 1 (PAI 1). This combination produces a paradoxical state of relative hypofibrinolysis, despite high tPA levels. α2 antiplasmin generally remains normal, but in the context of low plasminogen and high PAI 1, fibrin breakdown is inefficient [81]. In the study by Zanetto et al., 2021, HCC patients with cirrhosis had significantly lower plasmin–antiplasmin (PAP) complex levels than cirrhotic controls without HCC, reflecting reduced fibrinolytic activation despite similar individual factor levels [38]. The thrombin rich environment in HCC promotes thrombin-activatable fibrinolysis inhibitor (TAFI) activation, generating relative fibrinolytic resistance. Elevated TAFI levels are linked to sinusoidal microthrombosis, delayed clot lysis, and higher PVT risk, particularly in cirrhotic patients with large or multifocal tumors [82].

Progressive cirrhosis severity is characterized by declining hepatic derived factors (plasminogen, FXIII) and increasing extrahepatic factors (tPA, TAFI), driving a hypofibrinolytic, prothrombotic state. Fibrin degradation products and D dimers indicate ongoing low-grade coagulation with secondary fibrinolysis, yet remain insufficient to prevent portal vein thrombus propagation [38,81].

### 3.5. Hemodynamic Factors and Endothelial Factors

Endothelial homeostasis in cirrhosis and HCC is governed by a delicate interplay of vasoactive, angiogenic, and antithrombotic mediators. Portal hypertension initiates endothelial dysfunction and chronic inflammation, promoting vascular injury and predisposing to PVT. In HCC, portal vein tumor thrombosis (PVTT) further reduces portal flow through direct vascular invasion, with flow velocity declining as thrombus burden increases. Involvement of the main portal trunk can lead to near complete flow cessation, aggravating portal hypertension and compromising hepatic perfusion. Doppler ultrasonography consistently demonstrates altered or diminished flow patterns in this setting [83,84].

At the molecular level, endothelin 1 (ET 1) modulates sinusoidal tone by stimulating stellate cell contraction via ET A receptors while promoting vasodilatory Akt/eNOS signaling through ET B under physiologic conditions. Activation of vascular endothelial growth factor receptor 2 (VEGFR 2)—independent of VEGFA alone—triggers PI3K dependent eNOS phosphorylation, enhancing nitric oxide (NO) production. Prostacyclin (PGI_2_) complements this vasodilatory effect and inhibits platelet aggregation. However, in cirrhosis oxidative stress reduces both NO and PGI_2_ bioavailability, fostering intrahepatic vasoconstriction. Angiopoietin 2 (Ang 2) destabilizes sinusoidal vessels and sensitizes the endothelium to inflammatory injury, while the loss of microRNA 126 impairs angiogenic balance and endothelial repair. Simultaneously, downregulation of thrombomodulin (TM) and the endothelial protein C receptor (EPCR) impairs the protein C anticoagulant pathway, tipping the hemostatic equilibrium toward thrombosis.

Biomarker studies reinforce the concept of early endothelial and platelet activation in this milieu. Annexin A5 levels are markedly elevated in patients with PVT, with a higher Annexin-to-platelet ratio reflecting disproportionate cellular injury relative to platelet availability. In cirrhotic patients without HCC, an Annexin A5 threshold of >4.61 ng/mL achieved an AUC of 0.882 for predicting thrombosis, while in those with HCC, the AUC was 0.867, highlighting its sensitivity as an early marker of prothrombotic imbalance [85].

HCC exacerbates endothelial dysfunction through the secretion of IL 6, TNF α, and VEGF, and by expressing TF, which interacts with endothelial protease-activated receptors (PARs) to initiate local coagulation [45]. Epigenetic remodeling contributes to this prothrombotic niche: hypomethylation of pro-tumorigenic genes such as FAM83D and TNFRSF10A enhances tumor cell migration and endothelial destabilization [86,87], while hypermethylation of tumor suppressor genes like SCAND3 reduces apoptotic clearance of neoplastic clones [88]. Non-coding RNAs amplify these effects; downregulation of miR 381 increases VEGFA expression, promoting angiogenesis and vascular permeability [89], while long non-coding RNAs (lncRNAs) targeting FOSL2, CDK5, and CD44v6 facilitate endothelial mesenchymal transition, matrix remodeling, and immune modulation [90,91].

### 3.6. Inflammatory and Oxidative Stress Factors

The prothrombotic phenotype in HCC is amplified by persistent low-grade inflammation within the tumor and systemic milieu. Tumor-derived cytokines such as IL 6 and TNF α upregulate TF expression on endothelial and immune cells, promoting extrinsic coagulation activation [92]. These cytokines also trigger leukocyte recruitment and platelet aggregation, completing a thrombo-inflammatory amplification loop. Activation of neutrophils via NF κB signaling increases P selectin expression, which reinforces platelet–endothelial interactions and induces E selectin and IL 8 expression, further sustaining inflammation and leukocyte adhesion, thus predisposing to thrombosis [45].

The complement system contributes via generation of C3a and C5a, which promote leukocyte recruitment, endothelial activation, and neutrophil extracellular trap (NET) formation—structures that serve as scaffolds for platelet adhesion and fibrin deposition, thus propagating intravascular fibrin accumulation [93,94]. High-mobility group box 1 (HMGB1), released by both platelets and HCC cells, further exacerbates endothelial injury and promotes thrombosis through toll like receptor (TLR)-mediated pathways [45].

At a post-transcriptional level, downregulation of miR 223—a neutrophil-derived miRNA internalized by hepatocytes that normally suppresses IL 6 and TAZ—facilitates tumor progression and augments thrombotic propensity through dysregulation of platelet activation [95,96].

Moreover, Kupffer cell dysfunction contributes to this prothrombotic microenvironment. In experimental models of acute liver inflammation, Kupffer cells initiate intrasinusoidal thrombosis via STAT1 dependent upregulation of TF in both Kupffer cells and sinusoidal endothelial cells [97]. Depletion of Kupffer cells attenuates TF induction, fibrin deposition, and liver injury, underscoring their central role in hepatic procoagulant activation. In chronic liver disease, Kupffer cells also mediate intrahepatic platelet accumulation, recruiting platelets to sinusoids and further amplifying local thrombotic risk [98].

### 3.7. Gut Microbiome

The gut microbiota also plays a pivotal role in modulating thrombotic risk in cirrhotic patients, particularly through mechanisms involving dysbiosis, bacterial translocation, and systemic inflammation. A shift from beneficial commensals such as Bacteroides to potentially pathogenic taxa like Pseudomonas, Enterobacteriaceae, Acinetobacter, and Gammaproteobacteria is frequently observed [45]. This imbalance contributes to endotoxemia, Toll-like receptor activation, and downstream inflammatory cascades. These processes, in turn, stimulate vWF release, platelet aggregation, and activation of procoagulant pathways.

According to Huang et al., a distinct microbial profile has been identified in cirrhotic patients with PVT, including enrichment of Akkermansia, Eubacterium hallii group, Fusicatenibacter, and Anaerostipes, as well as depletion of Bacteroides, Enterococcus, and Weissella. This microbial shift correlates with an elevated D-dimer and platelet counts, reflecting an active thrombotic state [99]. Moreover, Qi et al. associated an enrichment of LPS-producing Gram-negative taxa, such as Escherichia-Shigella and Klebsiella, with a depletion of beneficial short-chain fatty acid-producing genera, like Faecalibacterium and Ruminococcus, with elevated portal LPS levels and Factor VIII, contributing to local endothelial activation and hypercoagulability [100].

In HCC, microbial dysbiosis and bacterial translocation further amplify systemic inflammation, coagulation cascade activation, and fibrotic progression. However, not all data are concordant: a recent cross-sectional study in end-stage liver disease found no significant association between overall microbiome diversity or TMAO levels and the presence of PVT [101].

## 4. New Diagnostic Tools and Risk Stratification Methods

Traditional coagulation assays like the INR fail to accurately capture the multifaceted balance of hemostasis, as they primarily reflect deficiencies in procoagulant factors without accounting for compensatory losses in anticoagulant. Advanced diagnostic modalities such as ROTEM and TGA have significantly enhanced our ability to assess the complex hemostatic alterations in patients with HCC, frequently revealing preserved or even hypercoagulable global clotting profiles. ROTEM provides real-time viscoelastic measurements of clot formation, firmness, and lysis, capturing the mechanical integrity of clot dynamics. In cirrhotic patients, including those with HCC, ROTEM has been shown to detect prothrombotic alterations such as increased clot firmness and impaired fibrinolysis, correlating with a heightened risk of PVT [18,40,102]. Complementing this, TGA quantifies the biochemical capacity to generate thrombin, offering insight into the kinetics and magnitude of coagulation potential. TGA measures include lag time, peak thrombin, time to peak, and endogenous thrombin potential, collectively delineating the thrombin-generating profile of the plasma. Studies have demonstrated that thrombin generation is significantly elevated in patients with HCC reflecting a tumor-driven hypercoagulable state that contributes to both portal and systemic thrombotic complications [38,65,103]. Moreover, platelet-rich TGA has revealed that platelet count and function critically modulate thrombin generation in cirrhosis. Below a threshold of 100 × 10^9^/L, platelet-dependent thrombin formation markedly declines, underscoring the dual contribution of cellular and plasma components to the coagulation balance [104]. Similarly, thromboelastography with platelet mapping (TEG-PM) offers an additional functional perspective by quantifying the platelet contribution to clot strength. In a prospective study of patients with HCC-related cerebral infarction, TEG-PM revealed a significantly higher maximum amplitude (MA) compared with controls, reflecting increased clot firmness driven by platelet–fibrin interaction. An MA cut-off of 61.35 mm independently predicted HCC-related cerebral infarction (OR 1.71, 95% CI 1.30–2.25; *p* < 0.001) with high accuracy (AUC 0.875; sensitivity 89.5%; specificity 66.4%), and is correlated with AFP, neutrophil count, and D-dimer levels, indicating that TEG-PM can identify a hypercoagulable phenotype and help recognize HCC patients at high thrombotic risk [105].

Zanetto et al. have shown that HCC exacerbates thrombin generation while impairing fibrinolysis, positioning malignancy as an independent driver of thrombophilia beyond the cirrhotic milieu [38]. Both ROTEM and TGA surpass standard assays in sensitivity and clinical relevance, particularly when used in tandem to assess different dimensions of coagulation: structural integrity versus enzymatic potential. Emerging evidence supports the integrated use of ROTEM and TGA to guide clinical decision-making across a range of scenarios, including anticoagulation, transfusion planning, and perioperative management. This dual approach has been applied in settings such as liver transplantation and high-risk surgery, where combined analysis informs fibrinogen replacement strategies, reduces unnecessary transfusions, and predicts hemorrhagic or thrombotic events with greater accuracy than either test alone [106,107,108,109]. In HCC, where hemostatic disturbances are driven by both hepatic insufficiency and tumor-specific mechanisms, the integration of ROTEM and TGA enables a comprehensive, real-time evaluation of coagulation dynamics.

## 5. Anticoagulant Treatment and Prophylaxis in HCC Patients

### 5.1. Anticoagulant Treatment

Therapeutic anticoagulation in cirrhotic patients with HCC and thrombosis demonstrated favorable outcomes in non-malignant PVT (Table 2). Recanalization was achieved in 61.9% of patients at 6 months [25] and 50.7% overall [110,111], with significant improvements in one-year survival (e.g., 58.8% vs. 20% in patients with and without recanalization, respectively) [110]. Senzolo et al. reported a marked reduction in PVT progression (from 62.3% to 9.1%) and improved thrombosis control with anticoagulation [112]. However, in cases of PVTT, anticoagulation did not yield significant benefit on thrombus progression or overall survival [112,113].

In terms of safety, major bleeding events ranged from 4.7% to 25%, depending on the study and patient population. Benevento et al. reported bleeding in 13.3% of HCC patients [110], while Kais et al. noted a 25% bleeding rate in anticoagulated patients [114]. In contrast, Senzolo et al. reported no bleeding events or treatment discontinuations among anticoagulated individuals [111]. These findings are complemented by a recent systematic review and meta-analysis that evaluated the safety of anticoagulation in patients with SVT and a history of portal hypertension (PH)-related bleeding. In this high-risk population, the cumulative incidence of PH-related rebleeding was lower among anticoagulated patients (17.1%) compared with untreated controls (40.0%). Observational studies showed a significantly reduced risk of rebleeding with anticoagulation (OR 0.15, 95% CI 0.04–0.52), while randomized controlled trials demonstrated a nonsignificant trend in the same direction (OR 0.84, 95% CI 0.31–2.32) [115]. Most adverse events were related to underlying hepatic dysfunction rather than the type of anticoagulant used.

Risk factors for poor outcomes and bleeding included advanced, older age, elevated bilirubin, and the presence of malignancy. Semmler et al. specifically identified Child–Pugh class B/C as an independent predictor of major spontaneous bleeding (aSHR 4.12) [116]. Overall, anticoagulation appears beneficial in selected cases of non-malignant PVT but requires cautious use in patients with advanced liver disease.

### 5.2. Primary Prophylaxis

Prophylactic anticoagulation in HCC remains a clinical challenge. To mitigate thrombotic risk, the oncology community has adopted predictive models such as the Khorana score, which incorporates tumor type, hemoglobin levels, leukocyte and platelet counts, and body mass index to stratify ambulatory cancer patients by thrombotic risk [5]. Pharmacologic prophylaxis is now recommended by the American Society of Clinical Oncology (ASCO) for hospitalized cancer patients and for ambulatory patients receiving systemic therapy with a Khorana score ≥ 2 [117]. However, these recommendations do not specifically address patients with HCC, who represent a unique subset due to the complex interplay between cancer-associated coagulopathy and the hemostatic alterations of underlying cirrhosis.

Despite this clear thrombotic propensity, prophylactic anticoagulation is still underused in HCC, largely due to longstanding fears of bleeding in cirrhotic patients. Traditionally considered “auto-anticoagulated,” cirrhotic individuals were thought to be at high bleeding risk due to elevated INR and thrombocytopenia. However, this concept has been refuted by the paradigm of “rebalanced hemostasis,” which recognizes that both procoagulant and anticoagulant pathways are simultaneously altered, and that the net hemostatic state may shift dynamically toward either thrombosis or bleeding, particularly in the presence of triggers such as infection, inflammation, or malignancy [38].

Primary prophylaxis of PVT has been explored in cirrhotic patients without malignancy, using both LMWH and DOACs. In a single-center RCT, enoxaparin prevented PVT and delayed decompensation in advanced cirrhosis. At week 48, PVT developed in 0% of treated patients vs. 16.6% of controls; at week 96, the incidence was 0% vs. 27.7%, improving survival without excess bleeding [118]. More recently, a multicenter, double-blind trial evaluated rivaroxaban 10 mg qd for 24 months in patients with cirrhosis and CSPH, Child–Pugh B7–C10. The primary endpoint of decompensation or death/LT occurred in 26.8% of rivaroxaban vs. 46.9% of placebo (HR 0.51, 95% CI 0.25–1.05); benefit was most pronounced in Child–Pugh B7 (HR 0.26, 95% CI 0.07–0.90), with no significant increase in major bleeding [119].

Only data about primary prophylaxis with LMWH in cirrhotic patients with HCC come from cohorts of cirrhotic patients undergoing hepatobiliary surgery resulted in low VTE incidence and good clinical outcomes (Table 2). Serrano et al. observed cumulative VTE rates of 0.35%, 2.5%, and 7.2% at 1, 3, and 6 months, respectively, with a 6.6% overall mortality [120]. Vivarelli et al. reported a 0.63% rate of postoperative VTE in the LMWH treated group versus 1.38% in controls [121]. Zhang et al. registered an incidence of VTE of 4.8% vs. 14.9% (OR 0.31, 95% CI 0.08–1.16; *p* = 0.091) in LMWH group, indicating a trend toward reduced thrombotic events without reaching statistical significance [122]. Kim et al. found no cases of VTE (DVT or PE) at 90 days following extended LMWH prophylaxis [123]. The safety profile was generally favorable. Bleeding complications were rare or minor: Kim et al. noted only two minor bleeding events (1.6%) with no need for transfusion or reoperation [123]. Vivarelli et al. reported a 3.18% incidence of postoperative hemorrhage in the LMWH group [120], comparable to controls. Zhang et al. showed no significant difference in bleeding events in LMWH group (3.2% vs. 9.5%, *p* = 0.298) [122]. The only clearly identified bleeding risk factor was the presence of esophageal varices, as reported by Vivarelli et al. [120], and major hepatectomy and transfusion, according to Zhang et al. [122]. No increased bleeding was associated with LMWH in patients without high-risk varices.

### 5.3. Secondary Prophylaxis

Evidence on efficacy and safety of anticoagulation in cirrhosis has become so compelling that major societies have issued guidelines. Recent ISTH recommendations endorse anticoagulation for symptomatic PVT and suggest it for asymptomatic, progressing PVT, marking a shift from caution to active prophylaxis in cirrhotic patients [124].

Secondary prophylaxis and comparative studies highlighted the efficacy of anticoagulation in reducing VTE recurrence and mortality (Table 2). Davis et al. found similar VTE recurrence between patients on VKA (12.2%) and DOACs (11.1%) [125]. Bikdeli et al. showed higher VTE recurrence in cirrhotic patients (HR 2.08), but similar fatal PE rates (0.5%) [126]. Candeloro et al. reported that anticoagulation significantly reduced VTE (HR 0.42), mortality (HR 0.23), and even major bleeding (HR 0.47), especially in cancer patients, where mortality dropped from 23.3 to 5.7 per 100 person-years [127].

Safety findings were variable. DOACs were generally better tolerated than VKA: Davis et al. reported 13.4% major bleeding in the VKA group vs. 7.4% with DOACs [125]. In contrast, Semmler et al. observed high bleeding rates with DOACs—31.7% overall and 17.3% major bleeding—which were more frequent in patients with advanced cirrhosis [116]. The RIETE registry showed higher fatal bleeding in cirrhotic patients (2.1% vs. 0.2%), despite similar overall bleeding rates [126].

The most consistent predictors of bleeding or poor outcomes were Child–Pugh class B/C liver disease and possibly the use of VKA in decompensated patients. Semmler et al. confirmed the independent association between liver dysfunction and bleeding, not HCC per se [116].

Notably, despite promising data on DOACs, most of the available evidence derives from retrospective observational studies with intrinsic limitations or from interventional trials in which patients with HCC were poorly represented. In the recently published API-CAT trial, which compared apixaban 5 mg vs. 2.5 mg bid for extended secondary prophylaxis of cancer-associated VTE, only 28 patients had hepatobiliary cancers (18 in the full-dose group and 10 in the reduced-dose group). This extremely small sample size precludes any robust conclusions for this population, although, overall, the trial confirmed the noninferiority of reduced-dose apixaban for prevention of recurrent VTE (2.1% vs. 2.8%) and showed a significant reduction in clinically relevant bleeding (12.1% vs. 15.6%) [128].

**Table 2 cancers-17-03413-t002:** Anticoagulant treatment; primary and secondary prophylaxis of PVT and VTE in HCC patients.

Study	Design	Year	Thrombosis Site	Population	HCC Treatment	AC Treatment or Prophylaxis	Outcome	Safety
**AC primary prophylaxis**								
Vivarelli et al. [121]	Retrospective	2010	VTE	229 patients with cirrhosis and HCC	Surgery	157 (68.5%) in LMWH in primary prophylaxis.	Incidence of VTE 0.63% in AC vs. 1.38% in control group (*p* = 0.38).	Postoperative hemorrhage 3.18% in AC vs. 1.38% in control group. Esophageal varices were linked to an increased risk of bleeding.
Kim BJ et al. [123]	Prospective	2017	VTE	124 patients undergoing liver surgery for malignancy (10, 8.1% HCC)	Surgery	LMWH extended primary prophylaxis (up to 28 days)	Extended pharmacologic thromboprophylaxis resulted in 0 cases of VTE (DVT or PE) during 90-day follow-up.	Only 2 patients (1.6%) had minor bleeding, with no transfusion or reoperation required.
P. Serrano et al. [120]	Prospective	2018	VTE	284 patients, 97 (34.2%) patients undergoing laparotomy for hepatobiliary cancer	Surgical, unknown number of HCC patients	LMWH, primary prophylaxis	VTE incidence 0.35% at 1, 2.5% at 3, and 7.2% at 6 months.	Mortality 6.6%. No bleeding analysis.
Zhang et al. [122]	Prospective	2025	VTE	140 patients undergoing surgery for HCC	Surgical	LMWH, primary prophylaxis	Incidence of VTE was 4.8% (3/62) vs. 14.9% (11/74) (*p* = 0.091, OR = 0.31, 95% CI: 0.08–1.16) in the group with reduced prophylaxis vs. no prophylaxis.	Reduced prophylaxis will not increase risk of hemorrhagic complications (OR = 0.34, 95% CI: 0.07–1.69, *p* = 0.186).
**AC treatment and secondary prophylaxis**								
Rajani et al. [129]	Retrospective	2010	PVT	173 patients with PVT, 40% (80) with cirrhosis, 27% (47) with malignancy	Unknown treatment	65% of patients in AC treatment with LMWH or VKA	No improvement in survival with anticoagulation therapy.	2 patients who had not received AC died of AVB.
Senzolo et al. [25]	Prosective	2018	SVT	149 patients with liver cirrhosis and SVT, 39 (26.2%) HCC	Unknown treatment	Treatment with LMWH or fondaparinux	Anticoagulation led to 61.9% recanalization at 6 months. One-year survival was higher with recanalization (58.8% vs. 20%, *p* = 0.03). No thrombotic recurrence or progression occurred.	Major bleeding was rare (4.7%)
Davis KA et al. [125]	Retrospective	2019	VTE	82 cirrhotic patients with VTE in VKA (26.8% with malignancy) and 27 in DOAC (25.9% with malignancy)	Unknown treatment	VKA vs. DOAC, treatment or secondary prophylaxis	Recurrent VTE occurred in 10 (12.2%) patients receiving treatment with warfarin and 3 (11.1%) of patients treated with DOAC therapy.	Eleven (13.4%) patients in the warfarin group experienced a major bleed compared to 2 (7.4%) of patients receiving DOAC therapy (*p* = 0.51)
Bikdeli et al. [126]	Prospective (RIETE registry)	2019	VTE	Among 43,611 patients with VTE,187 patients with cirrhosis, and 75 patients (40.1%) with active cancer	Unknown treatment	98.4% on AC (LMWH for most patients) for a median of 109 days	VTE recurrence higher in patients with cirrhosis (HR 2.08). Fatal PE rate was comparable (0.5% in both groups).	Overall bleeding comparable, fatal bleeding higher in patients with cirrhosis (2.1% vs. 0.2%).
Mahmoudi T et al. [112]	Retrospective	2019	PVT	51 patients with HCC and PVT	Surgery, LRTs or Systemic treatment	12 (23.5%) on AC vs. 39 (76.5%) without AC. VKA or LMWH treatment.	PVT progression: 50% with AC, 49%. AC had no impact (HR 1.32, 95% CI 0.41–4.19).	AC therapy carried a bleeding risk, suggesting the need for individualized treatment decisions.
Chen et al. [34]	Retrospective	2021	DVT	355 patients with HCC, 66 (18.6%) with DVT	Surgery	66 (100%) on AC treatment (unspecified)	DVT disappeared within 2 months in 63 (95.5%) cases, with chronic DVT remaining in 3 (4.5%) cases.	No PE or major bleeding occurred during the treatment.
Semmler et al. [116]	Retrospective	2021	VTE	33 patients with cirrhosis and HCC	48.5% on TKI treatment	All on DOACs	Thrombotic events and outcomes not specifically reported.	Bleeding occurred in 31.7%, major in 17.3%; associated with CPS B/C (aSHR: 4.12, *p* < 0.001), not with HCC.
Candeloro et al. [127]	Meta-analysis	2022	SVT (mostly PVT)	1635 patients with SVT, 278 (17%) with liver cirrhosis, 523 (32%) with solid cancer	Unknown treatment	None vs. AC (Heparin, VKA, or DOAC)	Anticoagulation reduced SVT (HR 0.42), bleeding (HR 0.47), and mortality (HR 0.23). In cancer patients, mortality dropped from 23.3 to 5.7 per 100 p-y (HR 8.68).	Among cirrhotic patients, bleeding risk was higher (HR 1.92) but outweighed by benefits.
Benevento et al. [110]	Retrospective	2023	PVT	162 patients with cirrhosis and PVT, 30 with HCC	Unknown treatment	LMWH (30.9%), VKA (6.2%), fondaparinux (5.5%); no DOACs used	Recanalization in 50.7% of treated patients; similar rates between HCC (46.1%) and non-HCC (51.9%); median time to recanalization: 4.5 months	Bleeding events: 13.3% in HCC vs. 25% in non-HCC (*p* = 0.230); treatment discontinuation due to bleeding: ~3% in both groups; no significant safety concerns.
Kais et al. [114]	Retrospective	2023	PVT	122 cirrhotic patients with HCC	Unknown treatment	54% anticoagulant (apixaban 56%, LMWH 21%, others 23%)	No survival benefit: median OS 6 months in both groups (HR 0.91, *p* = 0.72). Recanalization: not reported.	25% bleeding complications in the AC group.
Senzolo et al. [111]	Retrospective	2024	PVT	88 cirrhotic patients with HCC and PVT	Mixed, but mostly LPS-ablation	36.5% (22/83) PVT patients received anticoagulation (type/dose not specified)	PVT is associated with lower OS; anticoagulation significantly improved PVT outcomes (50% vs. 6.6% improvement; 9.1% vs. 62.3% progression).	No bleeding events or treatment discontinuations occurred.
Balcar et al. [113]	Retrospective	2024	PVT	124 cirrhotic patients with HCC, 47 (38%) with PVTT and 49 individuals (40%) with non-tumorous PVT	94 patients (76%) were treated with effective systemic therapies	24 individuals (19%) received therapeutic anticoagulation	AC did not affect malignant thrombosis. Systemic therapy (aHR 0.26) but no-AC was independently associated with reduced all-cause mortality.	Non-selective beta-blockers were associated with reduced risk of variceal bleeding or death from any cause (aHR 0.69).

Abbreviations. AC: Anticoagulation; VTE: Venous Thromboembolism; DVT: Deep Vein Thrombosis; PE: Pulmonary Embolism; PVT: Portal Vein Thrombosis; SVT: Splanchnic Vein Thrombosis; HCC: Hepatocellular Carcinoma; LMWH: Low-Molecular-Weight Heparin; VKA: Vitamin K Antagonist; DOAC: Direct Oral Anticoagulant; TKI: Tyrosine Kinase Inhibitor; OS: Overall Survival; LRT: Locoregional Therapy; LPS: Laparoscopic Surgery; AVB: Acute Variceal Bleeding; CPS: Child-Pugh Score; aSHR: Adjusted Subdistribution Hazard Ratio; HR: Hazard Ratio; CI: Confidence Interval; p-y: Person-Years.

## 6. Combination Therapies: Anticoagulants and Immunotherapy

The advent of immune checkpoint inhibitors, particularly the combination of atezolizumab and bevacizumab (AtezoBev), has reshaped the therapeutic landscape for patients with unresectable HCC. However, this regimen introduces distinct hemostatic challenges, largely attributable to bevacizumab’s inhibition of VEGF. VEGF blockade impairs angiogenesis and vascular remodeling, mechanisms crucial in cirrhotic patients with portal hypertension, thereby predisposing to both thrombotic and hemorrhagic complications, including variceal bleeding and VTE [103,130,131,132].

The pathophysiological basis lies in compromised vascular integrity and repair, which can exacerbate bleeding in patients with pre-existing coagulopathy or fragile varices, underscoring the importance of pre-treatment variceal screening and the optimization of portal hypertension management, including non-selective beta-blockers and endoscopic ligation when indicated [133].

Real-world evidence (Table 3) suggests that thromboembolic events in patients with HCC receiving AtezoBev are relatively uncommon and comparable to those seen with tyrosine kinase inhibitors (TKIs). In Khaled et al., thromboembolic events occurred in 6% of patients treated with AtezoBev compared with 4% in the lenvatinib cohort, indicating comparable thrombotic risk between the two regimens. Among the 95 patients (29%) receiving anticoagulation (60 LMWH, 3 VKA, 30 DOACs), thromboembolic rates were comparable to those in non-anticoagulated patients [103].

The use of therapeutic anticoagulation (AC), including DOACs, LMWH, and VKAs, appears feasible in this setting. Allaire et al. reported that 20% of patients were on curative AC without significant impact on mortality (aHR 0.75) [134]. Moriguchi et al. and Nakabori et al. confirmed that AC, including DOACs, did not adversely affect progression-free or overall survival, and thromboembolic events remained comparable (HR 1.357, *p* = 0.770) [135,136].

Bleeding complications remain a more prominent clinical concern, with overall rates ranging from 3% to 18%, depending on patient selection, underlying liver function, and anticoagulation use. Grade ≥ 3 bleeding occurred in approximately 7.9% of anticoagulated patients versus 10.7% in non-anticoagulated patients in the Moriguchi et al. cohort, with no fatal hemorrhages reported [134]. In Khaled et al., overall bleeding occurred in 18% of patients on AtezoBev versus 11% on Lenvatinib, with 3% variceal and 2.5% non-variceal gastrointestinal bleeding [103]. Acute variceal bleeding (AVB) was observed in 3–3.5% of cases in the Stefanini and Khaled studies [32,103], whereas Larrey et al. reported a higher 14% AVB incidence with a median onset of 3 months, again without bleeding-related mortality [137]. Allaire et al. reported a 12% 1-year AVB incidence in patients on AVB prophylaxis, with anticoagulation not significantly increasing mortality (aHR 0.75) or AVB risk (aHR 1.34) [134]. Nakabori et al. similarly found no significant difference in bleeding between patients treated with DOACs and those without anticoagulation (HR 1.357, *p* = 0.770) [136].

Across studies, independent predictors of bleeding included prior AVB (HR 10.58 and aHR 4.32) [137], large spleen size (OR 1.1, *p* = 0.03) [103], high ALBI score (HR 9.083, *p* = 0.039) [136], low serum albumin (HR 0.298, *p* = 0.023) [135], and PVTT (aHR 3.25) [134], whereas macrovascular invasion and ALBI grade >1 were associated with poorer overall prognosis [32].

**Table 3 cancers-17-03413-t003:** Safety of anticoagulant treatment in HCC patients undergoing AtezoBev.

Study	Design	Year	Thrombosis Site	Population	Anticoagulant Treatment or Prophylaxis	Safety	Risk Factors for Bleeding
Larrey et al. [137]	Prospective	2022	PVT + DVT/PE	43 patients with HCC undergoing AtezoBev	48.8% on curative AC (unspecified)	AVB incidence was higher with atezolizumab–bevacizumab than sorafenib (21% vs. 5% at 1 year, *p* = 0.02). No bleeding-related death.	- History of AVB (HR: 10.58, *p* = 0.03).
Moriguchi M. et al. [135]	Retrospective	2023	PVT + DVT/PE	185 patients (IMBRAVE150 in: 157; IMBRAVE150 out: 28)	Curative anticoagulants (unspecified) or antiplatelets	14 had grade ≥ 3 hemorrhage complications: 11 in the anticoagulant group (7.9%) and 3 in the other group (10.7%); no deaths attributable to bleeding events.	- High ALBI score. No significant PFS/OS difference.
Ben Khaled et al. [103]	Retrospective	2024	PVT + DVT/PE	325 patients with HCC undergoing AtezoBev vs. 139 undergoing Lenvatinib	95 (29%) patients in AC (60 LMWH, 3 VKA, 30 DOACs)	3 months bleeding in 18% of patients on AtezoBev and 11% on Lenv, variceal hemorrhage in 3% for both, and VTE events in 6% vs. 4%, not significant.	For GI bleeding: - Spleen size (OR: 1.1, *p* = 0.03) - History of variceal bleeding (OR: 3.0, *p* = 0.04). For non-GI bleeding: - anticoagulation use (OR: 2.2, *p* = 0.04).
Stefanini et al. [32]	Prospective	2024	PVT + DVT/PE	397 patients with HCC undergoing AtezoBev	9 patients on AC (unknown drugs)	Variceal bleeding (3.5%), digestive non-variceal bleeding (2.5%).	- PS > 0 - PVTT- AFP > 400 ng/mL, - ALBI grade > 1, - N/L ratio > 3.
Allaire et al. [134]	Prospective	2024	PVT + DVT/PE	200 patients with HCC undergoing AtezoBev on AVB prophylaxis	20% on curative AC	AVB incidence: 12% at 12 months; AC not associated with mortality (aHR: 0.75) or AVB (aHR: 1.34).	- PVTT (aHR: 3.25) - history of AVB < 6 months (aHR: 4.32) - varices of any size (aHR: 3.22).
Nakabori et al. [136]	Retrospective	2024	PVT + DVT/PE	59 patients with HCC undergoing AtezoBev	5 (8.5%) patients on DOAC	Bleeding rates did not differ between DOAC and no-AC groups; DOAC use was not associated with bleeding (HR: 1.357, *p* = 0.770).	- Low albumin (HR: 0.298, *p* = 0.023) - high ALBI score (HR: 9.083, *p* = 0.039).

Abbreviations. PVT: Portal Vein Thrombosis; DVT: Deep Vein Thrombosis; PE: Pulmonary Embolism; HCC: Hepatocellular Carcinoma; AC: Anticoagulation; LMWH: Low-Molecular-Weight Heparin; VKA: Vitamin K Antagonist; DOAC: Direct Oral Anticoagulant; AVB: Acute Variceal Bleeding; GI: Gastrointestinal; ALBI: Albumin–Bilirubin; PFS: Progression-Free Survival; OS: Overall Survival; OR: Odds Ratio; HR: Hazard Ratio; aHR: Adjusted Hazard Ratio; ECOG PS: Eastern Cooperative Oncology Group Performance Status; MVI: Macrovascular Invasion; AFP: Alpha-Fetoprotein.

## 7. Conclusions

Anticoagulation in cirrhosis is now widely recognized as both safe and effective, challenging the traditional fear of bleeding that has historically limited its use. Updated AGA, ISTH and EASL guidelines support anticoagulation for non-malignant PVT, discourage routine INR or platelet correction before procedures, and align with evidence from other cohorts showing that well-selected cirrhotic patients have comparable rates of major bleeding than untreated controls. Reduced-dose DOACs, particularly apixaban, have demonstrated favorable safety and efficacy in Child–Pugh B patients, solidifying a risk-adapted and proactive approach to thromboprophylaxis in this population.

In HCC, thrombosis risk is amplified by the interplay of malignancy-related and cirrhosis-driven procoagulant mechanisms. PVT complicates locoregional therapies, worsens transplant outcomes, and reduces overall survival, all of which highlight the value of early risk stratification and targeted anticoagulation. Evidence increasingly supports that anticoagulation in HCC patients, when individualized, does not result in unmanageable bleeding risk and can mitigate portal hypertension while potentially improving survival. Importantly, major bleeding rates in most series were acceptable and often attributable more to advanced liver dysfunction than to anticoagulant choice, underscoring the importance of careful patient selection.

Primary prophylaxis in HCC remains underexplored. Data from cirrhotic cohorts without malignancy suggest that LMWH and DOACs can effectively prevent PVT and delay decompensation, without excess bleeding, even in advanced cirrhosis. Surgical series in HCC patients also support the feasibility and safety of LMWH prophylaxis, provided that high-risk varices are excluded or adequately treated. Secondary prophylaxis demonstrates clear benefit in reducing VTE recurrence and mortality, with DOACs generally showing more favorable safety profiles than VKAs, although advanced Child–Pugh class consistently emerges as the strongest predictor of bleeding.

The scenario is more complex in patients treated with AtezoBev, where immune checkpoint blockade intersects with VEGF inhibition and cirrhosis-associated hemostatic fragility. Bevacizumab heightens the risk of non-gastrointestinal bleeding, particularly in the presence of varices or uncontrolled portal hypertension, yet these patients also face a high thrombotic burden, including PVT progression. Current evidence suggests that therapeutic anticoagulation—including DOACs, LMWH, and VKAs—is feasible and does not worsen survival or progression-free outcomes in this setting, although bleeding remains the main concern. Predictors of hemorrhage include prior variceal bleeding, large spleen size, poor hepatic function (high ALBI, low albumin), and PVTT, suggesting that individualized prophylaxis strategies should integrate both clinical and hemodynamic parameters.

Overall, a personalized anticoagulation strategy is warranted in HCC, balancing thrombosis prevention with bleeding risk in the context of cirrhosis and evolving systemic therapies. Dynamic risk stratification tools such as TGA and ROTEM, combined with standard predictors (Child–Pugh, ALBI, variceal status), may help refine patient selection. Future prospective trials specifically addressing anticoagulation in HCC, particularly in the immunotherapy era, are needed to establish robust, evidence-based guidelines. Until then, careful multidisciplinary evaluation remains key to safe and effective integration of anticoagulation into the therapeutic algorithm of HCC.

## Figures and Tables

**Figure 1 cancers-17-03413-f001:**
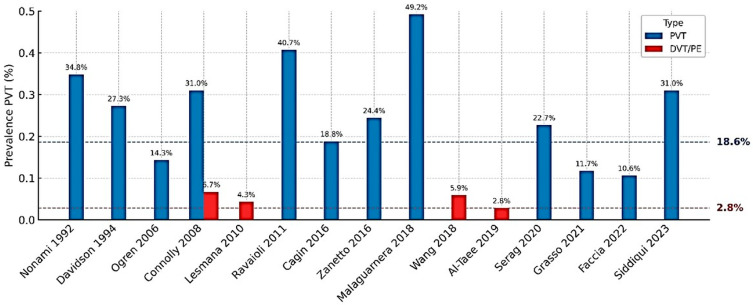
PVT and DVT/PE prevalence in HCC across studies, with weighted mean (blue for PVT, red for DVT/PE) [11,12,13,14,15,19,20,21,23,24,27,28,29,30,31].

**Figure 2 cancers-17-03413-f002:**
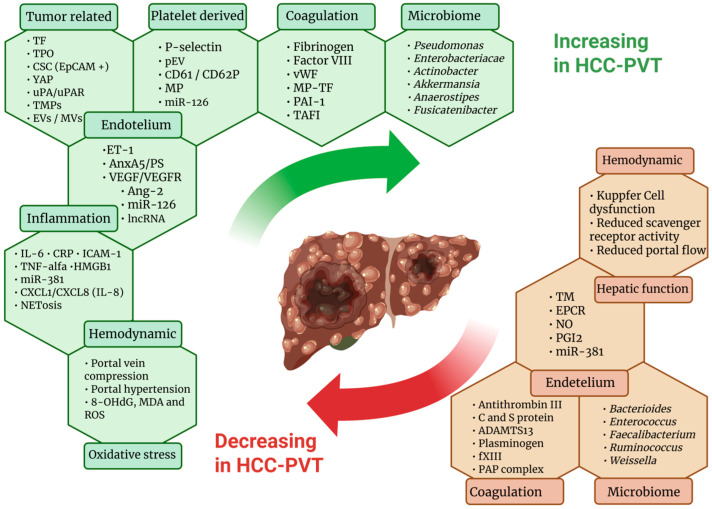
Molecular mechanisms of PVT and VTE in HCC. Created in Biorender. Abbreviations. TF: Tissue Factor; TPO: Thrombopoietin; CSC (EpCAM^+^): Cancer Stem Cells (Epithelial Cell Adhesion Molecule positive); YAP: Yes-associated protein; uPA/uPAR: Urokinase-type Plasminogen Activator/uPA Receptor; TMPs: Tumor-derived Microparticles; EVs/MVs: Extracellular Vesicles/Microvesicles; P-selectin: Platelet-selectin; pEV: Platelet-derived Extracellular Vesicle; CD61/CD62P: Cluster of Differentiation 61/62P; MP: Microparticles; miR-126: MicroRNA-126; Fibrinogen: Fibrinogen; Factor VIII: Coagulation Factor VIII; vWF: von Willebrand Factor; MP–TF: Microparticle–Tissue Factor; PAI-1: Plasminogen Activator Inhibitor-1; TAFI: Thrombin-Activatable Fibrinolysis Inhibitor; ET-1: Endothelin-1; AnxA5/PS: Annexin A5/Phosphatidylserine; VEGF/VEGFR: Vascular Endothelial Growth Factor/VEGF Receptor; Ang-2: Angiopoietin-2; lncRNA: Long Non-Coding RNA; IL-6: Interleukin-6; CRP: C-Reactive Protein; ICAM-1: Intercellular Adhesion Molecule-1; TNF-α: Tumor Necrosis Factor-alpha; HMGB1: High-Mobility Group Box 1; miR-381: MicroRNA-381; CXCL1/CXCL8 (IL-8): C-X-C Motif Chemokines 1 and 8 (Interleukin-8); NETosis: Neutrophil Extracellular Trap Formation; 8-OHdG: 8-Hydroxy-2’-deoxyguanosine; MDA: Malondialdehyde; ROS: Reactive Oxygen Species; TM: Thrombomodulin; EPCR: Endothelial Protein C Receptor; NO: Nitric Oxide; PGI_2_: Prostacyclin (Prostaglandin I_2_); Antithrombin III: Antithrombin III; Protein C/S: Protein C/Protein S; ADAMTS13: A Disintegrin And Metalloproteinase with Thrombospondin Motifs 13; Plasminogen: Plasminogen; fXIII: Coagulation Factor XIII; PAP complex: Plasmin–Antiplasmin Complex.

**Table 1 cancers-17-03413-t001:** PVT and VTE incidence and prevalence in HCC patients.

Study	Design	Year	Thrombotic Site	Population	HCC Treatment	Incidence/Prevalence
Nonami T. et al. [11]	Retrospective	1991	PVT	401 cirrhotic patients, 87 with HCC (21.7%)	OLT	15.7% PVT prevalence in cirrhosis without HCC. 34.8% prevalence in cirrhosis with HCC.
Davidson et al. [12]	Prospective	1994	PVT	132 patients with cirrhosis, 22 with HCC (16.7%)	OLT	27.3% (6/22) PVT prevalence in HCC vs. 9.1% (10/110) in non-HCC.
Ögren M et al. [14]	Retrospective	2006	PVT	392 HCC autoptic report, 182 with cirrhosis	No treatment	PVT prevalence 14.3% in cirrhosis with HCC (OR 17.1), 9.7% in HCC without (OR 5.2), and 4.5% in cirrhosis without HCC.
Connolly GC. et al. [15]	Retrospective	2008	PVT + DVT/PE	194 patients with cirrhosis and HCC	OLT or LRT	PVT: 31%; VTE: 6.7%; significantly impacting survival (mOS 2.3 m in PVT vs. 17.6 m without PVT); PVT patients with higher risk of VTE compared to patients. without PVT (11.5% vs. 4.4%; *p* = 0.04).
Lesmana et al. [30]	Case–control, single center	2010	DVT/PE	256 patients with cirrhosis, 87 with HCC	Unknown	4.6% (4/87) VTE prevalence without HCC vs. 4.3% (8/169) with HCC.
Rajani et al. [32]	Retrospective	2010	PVT	173 patients, 40% (80) with cirrhosis, 27% (47) with malignancy.	Unknown	PVT incidence rate was 0.7 per 100,000 per year and prevalence rate was 3.7 per 100,000 inhabitants.
Ravaioli et al. [13]	Retrospective	2011	PVT	889 patients with cirrhosis, 282 with HCC (31.7%)	OLT	PVT prevalence in cirrhosis with HCC is 40.7% vs. 30.7% in patients without HCC (*p* < 0.05).
Tzeng et al. [16]	Retrospective	2012	PVT + DVT/PE	5651 hepatectomies	Surgery	DVT (1.93%), PE (1.31%), venous thromboembolism (VTE) (2.88%).
Ponziani et al. [17]	Systematic review	2014	PVT	23,932 cirrhotic patients, (39%) with HCC	OLT	PVT incidence: 10.6% overall, 3.2/100 p-y (new), 7.4/100 p-y (total).
Cagin et al. [19]	Retrospective	2016	PVT	461 patients with cirrhosis, 69 (15%) with HCC	Unknown	9.8% (45/461) PVT overall prevalence vs. 18.8% (13/69) in HCC.
Zanetto et al. [31]	Prospective	2016	PVT + DVT/PE	76 patients with cirrhosis, 41 with HCC (53.9%)	Surgery, LRTs, systemic	24.4% (10/41) PVT incidence at 1 year in HCC vs. 11.4% (4/35) in non-HCC. Among HCC, 50% of PVT occurred in Child A. No DVT/PE
Wang Y et al. [20]	Retrospective	2018	PVT and DVT/PE	270 patients with cirrhosis and HCC	11.5% curative treatment, and 35.9% sorafenib	5.9% (16 cases) of VTE, including 7 (43.8%) pulmonary embolism, 4 (25%) peripheral deep vein thrombosis, and 6 (37.5%) intra-abdominal thrombosis.
Malaguarnera et al. [21]	Retrospective	2018	PVT	118 patients with liver cirrhosis and HCC	Unknown	49.2% (58 patients) PVT prevalence.
Senzolo et al. [25]	Review	2018	PVT	2721 cirrhotic patients, 41 with HCC	Unknown	~10% PVT prevalence is in CPA cirrhosis, 17% in CPB/C, and up to 26% in OLT candidates. 1-year incidence ranges from 3.7% to 24.4%, lower in cohorts with mostly compensated cirrhosis.
Violi et al. [22]	Prospective	2019	PVT	753 cirrhotic patients, 152 with HCC	Unknown	PVT incidence: 6.05/100 p-y (4.1 without prior PVT, 18.9 with prior PVT).
Al-Taee et al. [23]	Retrospective	2019	DVT/PE	54,275 hospitalized patients with HCC	Unknown	2.8% VTE prevalence (2.5% in 2008 to 3.0% in 2013, a significant increase).
Serag et al. [24]	Retrospective	2020	PVT	91 patients with cirrhosis, 44 (48.4%) with HCC	Unknown	12.7% (6/47) PVT prevalence in cirrhosis without HCC vs. 22.7% (10/44) in cirrhosis with HCC.
Schlick et al. [33]	Retrospective	2021	DVT/PE	Patients undergoing hepatectomy for malignancy (*n* = 11,172)	Hepatectomy (partial, left, right, trisegmentectomy)	Post-discharge VTE: 0.9% (overall VTE: 3.1%).
Ow et al. [26]	Retrospective	2021	PVT + DVT/PE	632 cirrhotic patients in ICU, 77 with HCC (12%)	Unknown	13% incidence of early VTE; 7.2% of late VTE (80% were PVT). No difference in survival between patients with and without VTE
Chen et al. [34]	Retrospective	2021	DVT	355 patients with HCC	Surgery	18.6% (66/355) DVT incidence after surgery.
Grasso et al. [27]	Retrospective	2021	PVT	750 patients with cirrhosis and HCC	LPS MWA	11.7% (88/750) prevalence of PVT. Cirrhosis severity, lack of response to HCC treatments, and complete/progressive PVT predictive of death.
Faccia et al. [28]	Retrospective	2022	PVT + DVT/PE	7445 cirrhotic patients accessing ED, 1524 with HCC	Unknown	In cirrhotic patients with PVT: 5.13% (382/7445), VTE: 1.27% (95/7445). Among HCC patients: PVT 10.63%, VTE 1.70%
Siddiqui et al. [29]	Retrospective cross-sectional analysis of prospective registry	2023	PVT	316 patients with HCC; subgroup analysis on Child–Pugh A)	All treatment allowed	PVT prevalence: 31% overall; 24.5% in Child–Pugh A;
Liu et al. [35]	Meta-analysis	2025	DVT/PE	935,639 patients with HCC	Unknown	VTE incidence in liver cancer was 35.85‰ overall, but higher with longer follow-up (up to 47.19‰) and markedly elevated in HCC with cirrhosis (229.56‰) vs. HCC alone (2.66‰).

Abbreviations. PVT: Portal Vein Thrombosis; DVT: Deep Vein Thrombosis; PE: Pulmonary Embolism; VTE: Venous Thromboembolism; HCC: Hepatocellular Carcinoma; OLT: Orthotopic Liver Transplantation; LRT: Locoregional Therapy; LPS: Laparoscopic Surgery; MWA: Microwave Ablation.

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
