# Peer review of "Thrombotic Risk and Coagulation Imbalance in Cirrhosis and Hepatocellular Carcinoma: Clinical Implications and Management"

_cancers, 2025, doi:10.3390/cancers17213413_

Round 1

Reviewer 1 Report

Comments and Suggestions for Authors

This is a great job, I have only some suggestions:

line 18:  The first abbreviation after the word shuld be preseented in the first presentation, e.g. rotational thromboelastography (ROTEM)

Line 150:  2.2= no mention about Venous Thromboembolic Events in Cancer target (TKI) and Immunotherapy, and the management

Line 192: the green is blue and violet is red in Figure (mismatch)
Line 769: Abbreviations =
This is a long paper, so please view it in alphabetical order to make it easier for readers.

Author Response

We sincerely thank the reviewer for the careful reading of our manuscript and for the insightful and constructive comments, which have greatly contributed to improving the clarity, focus, and overall quality of the paper.

line 18:  The first abbreviation after the word shuld be preseented in the first presentation, e.g. rotational thromboelastography (ROTEM)

Thank you for your observation. The first occurrence of each abbreviation, including rotational thromboelastography (ROTEM), has been corrected to ensure proper definition upon first mention.

Line 150:  2.2= no mention about Venous Thromboembolic Events in Cancer target (TKI) and Immunotherapy, and the management

We appreciate this insightful comment. Specific discussion of venous thromboembolic events associated with tyrosine kinase inhibitors (TKIs) and atezolizumab–bevacizumab combination therapy is included between lines 683 and 717. However, an in-depth analysis of these therapies lies beyond the primary scope of this review.

Line 192: the green is blue and violet is red in Figure (mismatch)

Thank you for pointing this out. The color mismatch in the figure (green/blue and violet/red) has been corrected accordingly.

Line 769: Abbreviations = This is a long paper, so please view it in alphabetical order to make it easier for readers.

We thank the reviewer for this helpful suggestion. The list of abbreviations has been reordered alphabetically to facilitate easier consultation by readers.

Reviewer 2 Report

Comments and Suggestions for Authors

This review covers a clinically important issue on thrombosis in HHC and cirrhosis. It is well-written and comprehensive and useful for the clinician

I have the following comments:

The title should be changed to ‘.. cirrhosis and hepatocellular carcinoma…’ since this covers the content better

Can the authors provide clear clinical recommendations for anticoagulation in HCC

Consistent use of abbreviations throughout the text must be ensured

Very fine and comprehensive tables

Author Response

We sincerely thank the reviewer for the careful reading of our manuscript and for the insightful and constructive comments, which have greatly contributed to improving the clarity, focus, and overall quality of the paper.

The title should be changed to ‘.. cirrhosis and hepatocellular carcinoma…’ since this covers the content better

Thank you for the valuable suggestion. The title has been revised to “... cirrhosis and hepatocellular carcinoma ...” to better reflect the scope and content of the manuscript.

Can the authors provide clear clinical recommendations for anticoagulation in HCC

We appreciate this important point. Current evidence remains insufficient to formulate clear, evidence-based recommendations. However, as stated in the conclusions, available data indicate that anticoagulant therapy, including direct oral anticoagulants, appears safe and feasible in patients with cirrhosis and HCC, when carefully selected.

Consistent use of abbreviations throughout the text must be ensured

Thank you for noting this. All abbreviations have been reviewed and standardized throughout the manuscript to ensure consistency and clarity.

Reviewer 3 Report

Comments and Suggestions for Authors

very good review.

  1. Line 141,430,701- you mention about tumor invasion of vasculature. Can you add few lines on differentiation of portal vein thrombosis with adjacent HCC and imaging ways to differentiate tumor thrombus from bland thrombus.
  2. You didnt mention Thromboelastography (TEG)  Platelet Mapping, can you add that too and explain the technique.Cen G, Song Y, Chen S, Liu L, Wang J, Zhang J, Li J, Li G, Li H, Liang H, Liang Z. The investigation on the hypercoagulability of hepatocellular carcinoma-related cerebral infarction with thromboelastography. Brain Behav. 2023 Apr;13(4):e2961. doi: 10.1002/brb3.2961. Epub 2023 Mar 16. PMID: 36929158; PMCID: PMC10097062.

Author Response

We sincerely thank the reviewer for the careful reading of our manuscript and for the insightful and constructive comments, which have greatly contributed to improving the clarity, focus, and overall quality of the paper.

Line 141,430,701- you mention about tumor invasion of vasculature. Can you add few lines on differentiation of portal vein thrombosis with adjacent HCC and imaging ways to differentiate tumor thrombus from bland thrombus.

We thank the reviewer for this thoughtful suggestion. A paragraph discussing imaging-based differentiation between bland portal vein thrombosis (PVT) and tumor thrombus (PVTT) was originally included in the early draft. However, it was later removed to improve focus and flow, as the primary aim of this review is to analyze thrombotic mechanisms in HCC and the efficacy and safety of anticoagulation and prophylaxis strategies. A detailed discussion of radiologic criteria and diagnostic differentiation between PVT and PVTT would require a dedicated review in itself and lies beyond the scope of the present manuscript.

You didnt mention Thromboelastography (TEG)  Platelet Mapping, can you add that too and explain the technique.Cen G, Song Y, Chen S, Liu L, Wang J, Zhang J, Li J, Li G, Li H, Liang H, Liang Z. The investigation on the hypercoagulability of hepatocellular carcinoma-related cerebral infarction with thromboelastography. Brain Behav. 2023 Apr;13(4):e2961. doi: 10.1002/brb3.2961. Epub 2023 Mar 16. PMID: 36929158; PMCID: PMC10097062.

We appreciate this excellent recommendation. A new section has been added discussing TEG with Platelet Mapping, including the quote suggested. The paragraph explains the technique and its potential role in identifying hypercoagulable phenotypes and high-risk patients with HCC.